# Screening Leads to Overestimated Associations of Thyroid Dysfunction and Thyroiditis with Thyroid Cancer Risk

**DOI:** 10.3390/cancers13215385

**Published:** 2021-10-27

**Authors:** Young Shin Song, Kyung Soo Kim, Soo Kyung Kim, Young Wook Cho, Hyo Geun Choi

**Affiliations:** 1Department of Internal Medicine, CHA Bundang Medical Center, CHA University, Seongnam 13496, Korea; yssongmd@gmail.com (Y.S.S.); kks982@hanmail.net (K.S.K.); imdrksk@chol.com (S.K.K.); ywcho@cha.ac.kr (Y.W.C.); 2Hallym Data Science Laboratory, Hallym University College of Medicine, Anyang 14068, Korea; 3Department of Otorhinolaryngology-Head & Neck Surgery, Hallym University College of Medicine, Anyang 14068, Korea

**Keywords:** thyroid cancer, hyperthyroidism, hypothyroidism, screening, nationwide studies, case-control studies

## Abstract

**Simple Summary:**

The association of thyroid dysfunction and thyroiditis with the risk of thyroid cancer is an important issue in clinical practice, and is controversial due to various confounders, including screening effects. In this large-sample, population-based, nationwide study, we demonstrated that the screening effect likely significantly contributed to the positive associations of thyroid dysfunction and thyroiditis with thyroid cancer. After adjustment for this confounder, thyroid cancer risk was positively associated with hypothyroidism and thyroiditis but negatively associated with hyperthyroidism and Graves’ disease. Multiple subgroup analyses showed consistent results. Given these overestimated associations, we suggest that unnecessary and excessive concerns or screening tests for thyroid cancer could be reduced in patients with thyroid dysfunction or thyroiditis.

**Abstract:**

We aimed to assess the relationships of functional thyroid disease and thyroiditis with subsequent thyroid cancer, which is controversial due to various confounders, and the effect of thyroid disease workup on this association. We used the cohort data from 2002 to 2015 (Study I, *n* = 28,330) and the entire data from 2002 to 2019 (Study II, *n* = 883,074) of the Korean National Health Insurance Service database, and performed logistic regression and subgroup analyses with various covariates. In Study I, hypothyroidism, thyroiditis, autoimmune thyroiditis, hyperthyroidism, and Graves’ disease showed positive associations with thyroid cancer. In Study II, after adjustment for covariates including the number of thyroid function tests, the ORs for thyroid cancer were significantly reduced in all thyroid diseases. Hypothyroidism, thyroiditis, and autoimmune thyroiditis were positively associated (adjusted odds ratio, OR (95% confidence interval, CI) 1.28 (1.25–1.32), 1.36 (1.31–1.42), and 1.17 (1.11–1.24), respectively), whereas hyperthyroidism and Graves’ disease were negatively associated with thyroid cancer (adjusted OR (95% CI) 0.80 (0.77–0.83) and 0.69 (0.65–0.74), respectively). Multiple subgroup analyses in both studies showed consistent results. In this large population-based, nationwide study, we confirmed that thyroid disease workup leads to overestimation of associations of thyroid dysfunction and thyroiditis with thyroid cancer risk.

## 1. Introduction

The associations between functional thyroid diseases and thyroid cancer are controversial. While some studies have reported that hypothyroidism increases the risk of thyroid cancer [1,2,3], others have reported conflicting results [4,5,6,7,8]. Some studies [3,4,8], though not all [5,6,7], have shown an increased risk of subsequent thyroid cancer in patients with hyperthyroidism. Thyroiditis, including autoimmune thyroiditis, such as Hashimoto’s thyroiditis and Graves’ disease, the most common causes of thyroid dysfunction in iodine-sufficient areas, has also been associated with thyroid cancer risk, but with low consistency [9,10,11].

Thyroid dysfunction and thyroiditis may be involved in the carcinogenesis of thyroid cancer. Thyroid hormone and thyroid-stimulating hormone (TSH) can enhance thyroid cancer cell proliferation [12,13] and angiogenesis [14,15] and regulate the expression of several genes [16,17,18]. Furthermore, thyroid dysfunction may have indirect effects through confounding factors such as obesity [19], serum cholesterol level [20], and smoking [21], which are known to be associated with thyroid cancer. The inflammatory condition of thyroiditis may cause DNA damage, resulting in mutations that lead to the development of thyroid cancer [22]. Moreover, Hashimoto’s thyroiditis might share developmental mechanisms with thyroid cancer, such as the activation of carcinogenic pathways [23] or stimulation by excessive iodine intake [24]. Graves’ disease can promote thyroid cancer growth via TSH-mimicking effects of the TSH-receptor stimulating antibody [25]. Additionally, there may be a screening effect, as patients with benign thyroid diseases may undergo more thyroid ultrasound examinations, leading to a higher thyroid cancer detection rate in this population than in those without thyroid disease.

Recently, a meta-analysis of 15 observational studies reported that hyperthyroidism was associated with an increased risk of thyroid cancer, and hypothyroidism was associated with an increased risk of thyroid cancer within the first 10 years after hypothyroidism diagnosis [3]. However, this analysis did not distinguish thyroiditis from thyroid dysfunction, and data on potential confounders were often lacking in most of the included studies.

In the current study, we hypothesized that functional thyroid disease and thyroiditis might be associated with an increased risk of thyroid cancer, and the screening effect could significantly contribute to the associations. Therefore, we investigated the associations of benign thyroid diseases with thyroid cancer using nationwide cohort data (Study I), and in order to determine whether these were real causal relationships or relationships due to increased detection, we evaluated the screening effect using nationwide data covering the entire population of Korea (Study II).

## 2. Materials and Methods

### 2.1. Ethics

The ethics committees of Hallym University (IRB number: 2019-10-023) and CHA Bundang Medical Center (IRB number: 2020-01-039) permitted this study. Written informed consent was waived by the Institutional Review Board. All analyses followed the guidelines and regulations of the ethics committee of Hallym University and CHA Bundang Medical Center.

### 2.2. Study Population and Participant Selection

This study was divided into Study I, which used Korean National Health Insurance Service (NHIS)-Health Screening Cohort data, consisting of a 10% random sample of all health screening participants [26,27], and Study II, which used NHIS data covering the entire population of South Korea [28].

In Study I, the cohort data for the years 2002 to 2015 were analyzed. Thyroid cancer patients were selected from 514,866 participants with 615,488,428 medical claim codes (*n* = 5769). The control group was selected from all participants who were not thyroid cancer patients (*n* = 509,097). To include only patients who were newly diagnosed with thyroid cancer, we excluded patients with thyroid cancer who were diagnosed in 2002 (*n* = 102). Among the thyroid cancer patients, a patient without total cholesterol data was excluded (*n* = 1). Among the control participants, we excluded those who died before 2003 or who were missing records after 2003 (*n* = 34) and those who had an International Classification of Diseases Revision 10 (ICD-10) code of C73 without thyroidectomy (*n* = 2054). Thyroid cancer patients were 1:4 matched with control participants for age, sex, income, and region of residence. To diminish selection bias, the control participants were chosen randomly with a random number method. The index date of each thyroid cancer patient was defined as the time of diagnosis of thyroid cancer. The index date of each control participant was determined as the index date of their matched thyroid cancer patient. During the matching process, 484,345 control participants were excluded. Conclusively, 5666 thyroid cancer patients were 1:4 matched with 22,664 control participants (Figure 1A).

In Study II, the NHIS provided a customized database for the years 2002 to 2019, which we had requested. Among the medical information of all the citizens who had enrolled in the national health insurance program, 441,537 thyroid cancer participants and 441,537 control participants were selected by 1:1 matching according to age, sex, and region of residence. In both the thyroid cancer and control groups, we excluded participants who did not have information on their region of residence (*n* = 21, respectively). Thyroid cancer patients were excluded if they did not undergo thyroidectomy (*n* = 198,775). Control participants were excluded if they were diagnosed with ICD-10 code C73 (*n* = 5249). We excluded the participants who did not have health check information (*n* = 24,100 thyroid cancer patients, and *n* = 78,236 controls). To include only thyroid cancer patients who were newly diagnosed, we excluded participants with thyroid cancer who were diagnosed in 2002 and 2003 (*n* = 5345). Eleven thyroid cancer patients were eliminated due to an error in the death date. Thyroid cancer patients were rematched with control participants in a 1:1 ratio according to age, sex, and region of residence due to the exclusion of participants. As in Study I, the index date for the thyroid cancer patients was defined as the date of thyroid cancer diagnosis, and the index date for the control participants was the index date of their matched thyroid cancer patients. During the matching process, 144,749 control participants and 3 thyroid cancer patients were excluded. Finally, 213,282 thyroid cancer patients were 1:1 matched with 213,282 control participants (Figure 1B).

### 2.3. Exposure (Thyroid Diseases)

(1)Hypothyroidism was defined as the presence of ICD-10 codes E02 (subclinical iodine-deficiency hypothyroidism) and E03 (other hypothyroidism). Among such patients, we selected patients with ≥2 records of these codes.(2)Hyperthyroidism was defined as the presence of ICD-10 code E05 (hyperthyroidism (thyrotoxicosis)). Among such patients, we selected patients with ≥2 records of this code.(3)Thyroiditis was defined as the presence of ICD-10 code E06 (thyroiditis). Among such patients, we selected patients with ≥2 records of this code.(4)Autoimmune thyroiditis was defined as the presence of ICD-10 code E063 (autoimmune thyroiditis). Among such patients, we selected patients with ≥2 records of this code.(5)Graves’ disease was defined as the presence of ICD-10 code E050 (thyrotoxicosis with diffuse goiter) and treatment with antithyroid medication ≥ 3 months in Study I, and the condition of treatment was not applied in Study II.

### 2.4. Outcome (Thyroid Cancer)

Thyroid cancer was confirmed if the patient was diagnosed with thyroid cancer (ICD-10 code C73). Among such patients, we selected only patients treated with thyroidectomy (claim codes P4551, P4552, P4553, P4554, and P4561), following our previous studies [29,30].

### 2.5. Covariates

Age groups were divided into 5-year intervals: 40–44, 45–49, …, and 85+ years old (total of 10 age groups) in Study I, and 21–25, 26–30, …, and 76+ years old (total of 12 age groups) in Study II. The region of residence was classified as urban or rural following our previous study [31]. Income groups were classified into 5 classes (class 1 (lowest income)–5 (highest income)). Tobacco smoking, alcohol consumption, and obesity, determined by body mass index (BMI, kg/m^2^), were categorized in the same ways as in our previous study [32]. Systolic blood pressure (SBP), diastolic blood pressure (DBP), fasting blood glucose, and total cholesterol were measured. The Charlson comorbidity index (CCI), which considers 17 comorbidities and has been widely used to measure disease burden, was a continuous variable (0 (no comorbidities) through 29 (multiple comorbidities)) [33]. Among the comorbidities, we excluded thyroid cancer. The number of thyroid function tests (claim codes: B5330, C3300, C3340, C3360, C7330, and C7334) was counted from 2 years before the index date to the index date in Study II.

### 2.6. Statistical Analyses

General characteristics were compared between the thyroid cancer and control groups using the chi-square test for categorical variables and the Wilcoxon rank-sum test for continuous variables, as the distribution of continuous variables was non-normal.

To estimate the odds ratios (ORs) with 95% confidence intervals (CIs) considering the sum of dates of each thyroid disease for thyroid cancer, logistic regression was used. In this analysis in Study I, the conditional logistic regression models were stratified by age, sex, region of residence, and income. A crude model, model 1 (adjusted for total cholesterol, SBP, DBP, fasting blood glucose, obesity, smoking, alcohol consumption, and CCI score), model 2 (model 1 plus hypothyroidism, hyperthyroidism, and thyroiditis), and model 3 (model 1 plus hypothyroidism, autoimmune thyroiditis, and Graves’ disease) were constructed. In Study II, instead of stratifying age, sex, region of residence, and income, we included these variables as covariates in the adjusted models because unconditional logistic regression was performed, and we added the number of thyroid function tests as a covariate. As autoimmune thyroiditis (E063) was included under thyroiditis (E06) and Graves’ disease (E050) was included under hyperthyroidism (E05), we constructed model 3 after calculating the ORs for hypothyroidism, hyperthyroidism, and thyroiditis in model 2.

For the subgroup analyses, we divided participants by age, sex, income, region, thyroid disease, obesity, smoking, alcohol consumption, CCI score, total cholesterol, blood pressure, and fasting blood glucose, and performed logistic regression.

Two-tailed analyses were performed, and significance was defined as a *p*-value less than 0.05. SAS version 9.4 (SAS Institute Inc., Cary, NC, USA) was used for the statistical analyses.

## 3. Results

### 3.1. Study I (Cohort Data from the NHIS)

The prevalence rates of all thyroid diseases, such as hypothyroidism, hyperthyroidism, thyroiditis, autoimmune thyroiditis, and Graves’ disease, were significantly higher in the thyroid cancer patients than in the controls (12.5% vs. 3.9% for hypothyroidism, 6.5% vs. 2.9% for hyperthyroidism, 6.0% vs. 2.0% for thyroiditis, 2.7% vs. 0.9% for autoimmune thyroiditis, and 0.7% vs. 0.3% for Graves’ disease; all *p* < 0.001; Table 1). Total cholesterol, SBP, DBP, obesity, smoking status, and the CCI score were different between the thyroid cancer group and the control group (*p* < 0.001).

The odds of previous hypothyroidism, hyperthyroidism, thyroiditis, autoimmune thyroiditis, and Graves’ disease in the thyroid cancer patients were significantly higher than those in the controls, even after adjustment for multiple possible confounding factors (Table 2). Among the thyroid diseases, the odds of hypothyroidism were highest (adjusted OR 3.05, 95% CI 2.73–3.41), followed by thyroiditis and autoimmune thyroiditis (adjusted OR 2.12, 95% CI 1.81–2.49 for thyroiditis, and adjusted OR 1.79, 95% CI 1.42–2.26 for autoimmune thyroiditis). Moreover, the thyroid cancer patients were 1.73 and 1.76 times more likely to have previous hyperthyroidism and Graves’ disease than the controls respectively, in the final model (95% CI 1.50–2.00 for hyperthyroidism, and 95% CI 1.17–2.66 for Graves’ disease).

Each thyroid disease was associated with high odds of thyroid cancer in the subgroup aged <60 and ≥60 years, although Graves’ disease showed a lack of statistical significance in model 3 due to the small number of patients (Figure 2A and Appendix A). Regarding sex, thyroid cancer was associated with all the thyroid diseases in the female subjects, and it was associated with hypothyroidism, hyperthyroidism, and thyroiditis in the male subjects. Interestingly, especially in the male subjects, the odds of previous hypothyroidism were markedly higher, at 8.63 times higher in the adjusted model, in the thyroid cancer patients than in the controls (95% CI 5.55–13.43, Appendix A). When stratified by income and region of residence, thyroid cancer patients were also more likely to have had thyroid diseases (Figure 2A). In addition, to exclude the influence of thyroid dysfunction and thyroiditis as confounding factors, we performed subgroup analyses according to the state of each thyroid disease (Figure 2B and Appendix A). Hypothyroidism and hyperthyroidism remained significantly associated with thyroid cancer in the groups without thyroiditis or autoimmune thyroiditis, and thyroiditis also remained significantly associated with increased odds in the groups without hypothyroidism or hyperthyroidism.

When we performed additional subgroup analyses according to obesity, smoking status, alcohol consumption, CCI score, total cholesterol, blood pressure, and fasting blood glucose, positive associations of each thyroid disease with thyroid cancer were demonstrated in most subgroups, except for some subgroups in which the number of patients was too small to show statistical significance (Figure 2C,D; Appendix A).

### 3.2. Study II (Entire Population Data from the NHIS)

In Study I, there were significant positive associations between each thyroid disease and thyroid cancer. However, to determine whether these were real causal relationships or due to ascertainment, in Study II, we performed further analyses using the entire dataset of the Korean NHIS, which was larger than the cohort dataset in Study I.

Most of the characteristics of the study subjects of Study II were similar to those of the subjects in Study I (Table 3). Nevertheless, the age groups of 21–40 years old were additionally included and not matched with controls for income, which could indirectly reflect the accessibility of healthcare services. The thyroid cancer patients had higher incomes than the controls (*p* < 0.001).

Patients who were diagnosed with any thyroid disease within one year before the index date accounted for 9.3% (19,837/213,282) and 0.9% (1814/213,282) of the thyroid cancer group and the matched control group respectively, and thyroid disease was approximately 11 times more prevalent in the thyroid cancer group than in the matched control group. Moreover, in all the thyroid disease groups, the duration from thyroid disease diagnosis to the index date was shorter in the thyroid cancer group than in the control group (43.6 ± 45.9 vs. 70.6 ± 47.3 months for hypothyroidism, 53.9 ± 50.7 vs. 79.3 ± 49.8 months for hyperthyroidism, 37.3 ± 45.2 vs. 66.7 ± 46.9 months for thyroiditis, 34.9 ± 43.4 vs. 63.8 ± 46.3 months for autoimmune thyroiditis, and 46.6 ± 48.3 vs. 75.9 ± 49.6 months for Graves’ disease). These findings suggested that detection bias could be a major potential confounder of the relationships between thyroid diseases and thyroid cancer.

Therefore, we attempted to overcome detection bias by adding the number of thyroid function tests as a covariate in the analysis models in Study II (Table 2). As in Study I, all thyroid diseases showed significant positive associations with thyroid cancer in the crude model. However, after adjustment for covariates, including the number of thyroid function tests, the OR values for thyroid cancer associated with all thyroid diseases were significantly reduced (unadjusted and adjusted ORs for thyroid cancer associated with each thyroid disease were as follows: 3.08 (95% CI, 3.00–3.16) and 1.28 (95% CI, 1.25–1.32) for hypothyroidism, 2.36 (95% CI, 2.29–2.43) and 0.80 (95% CI, 0.77–0.83) for hyperthyroidism, 3.19 (95% CI, 3.08–3.30) and 1.36 (95% CI, 1.31–1.42) for thyroiditis, 3.32 (95% CI, 3.15–3.49) and 1.17 (95% CI, 1.11–1.24) for autoimmune thyroiditis, and 2.44 (95% CI, 2.32–2.56) and 0.69 (95% CI, 0.65–0.74) for Graves’ disease). Notably, hyperthyroidism and Graves’ disease showed negative associations with thyroid cancer.

The subgroup analyses also confirmed that the thyroid cancer patients had higher rates of hypothyroidism, thyroiditis, and autoimmune thyroiditis and lower rates of hyperthyroidism and Graves’ disease (Figure 3 and Appendix A). Even after adjustment for the number of thyroid function tests, the increases in the odds of hypothyroidism and thyroiditis in male thyroid cancer patients were more prominent than those in the other subgroups (adjusted OR 2.22, 95% CI 1.93–2.56 for hypothyroidism, and adjusted OR 2.87, 95% CI 2.35–3.50 for thyroiditis, Figure 3A and Appendix A), consistent with Study I. Moreover, thyroiditis in past and current smokers was found to be associated with an approximately two times higher OR for thyroid cancer (adjusted OR 1.96, 95% CI 1.69–2.27, Figure 3C and Appendix A).

## 4. Discussion

In this large-sample, population-based, nationwide study, we found that thyroid cancer patients were two to three times more likely to have had previous thyroid diseases, such as hypothyroidism, hyperthyroidism, thyroiditis, autoimmune thyroiditis, or Graves’ disease, than the controls. However, when we added the number of thyroid function tests as a covariate to the analysis models to adjust for the effect of thyroid disease evaluation, the risk for thyroid cancer was significantly reduced in all thyroid diseases. Hypothyroidism, thyroiditis, and autoimmune thyroiditis remained positively associated, whereas hyperthyroidism and Graves’ disease were negatively associated with thyroid cancer. The subgroup analyses according to various covariates showed consistent results, and male thyroid cancer patients had markedly greater odds of previous hypothyroidism and thyroiditis than controls.

Unlike thyroid goiters and benign nodules, which have been demonstrated to have a relatively strong association with thyroid cancer, thyroid dysfunction has a weak association with thyroid cancer [4,5,6,7,8], and its role in thyroid cancer development remains unclear. In a previous pooled analysis of 12 case-control studies, there was no association between hypothyroidism and thyroid cancer [4]. Moreover, hyperthyroidism was marginally associated with subsequent thyroid cancer, but this association was not significant after two years of diagnosis of hyperthyroidism. In a very recent meta-analysis of 15 cohort and case-control studies, the associations between both types of thyroid dysfunction and thyroid cancer were relatively stronger than those in the prior pooled analysis [3]. Hypothyroidism was related to an increased risk of thyroid cancer (pooled risk ratio 3.31) up to 10 years after the hypothyroidism diagnosis, while hyperthyroidism was associated with a persistently increased risk of thyroid cancer (pooled risk ratio 4.49). However, a limitation of these observational studies is the presence of confounding factors, making it difficult to establish a clear causal relationship. In this study, the thyroid cancer patients had a significantly shorter average duration from the diagnosis of each thyroid disease to the index date of thyroid cancer than the controls, and a duration of less than one year was approximately 11 times more common in thyroid cancer patients than in controls. This indicates that screening for thyroid disease may be a major confounder. After adjustment for the number of thyroid function tests as a covariate, we found that screening had a significant effect on the relationship between thyroid disease and thyroid cancer risk. Given this overestimated relationship, unnecessary and excessive concerns or screening tests for thyroid cancer can be reduced in patients with thyroid dysfunction or thyroiditis.

The causal relationship between inflammation and cancer is widely accepted, and many cancers arise from sites of infection or chronic inflammation [34,35]. Thyroiditis and autoimmune thyroiditis can create a favorable microenvironment for the development and progression of thyroid cancer by various mechanisms, such as the increased production of reactive oxygen species, resulting in unstable chromatin conformation or DNA damage [36], and the activation of oncogenic signaling pathways [23,37]. Moreover, these diseases may coexist with thyroid dysfunction, which can affect the development and growth of thyroid cancer. Since Dailey et al. first reported the link between Hashimoto’s thyroiditis and papillary thyroid cancer in 1955 [38], there have been numerous studies to determine whether Hashimoto’s thyroiditis increases the risk of thyroid cancer, and the results are still controversial [10]. Thus, we analyzed thyroiditis and autoimmune thyroiditis by adjusting for the effects of thyroid dysfunction using logistic regression models and subgroup analyses, and we confirmed the associations of each thyroid disease with thyroid cancer.

Our study has several strengths. First, we analyzed large-sample, representative, nationwide population data. Not only cohort data but also entire NHIS data were used for validation. Second, we adjusted for the effect of thyroid disease workup, which is a major potential confounder, and confirmed its significant effect. In addition, to control for various other confounding factors, the participants in the control group were randomly selected and matched, and a number of subgroup analyses were performed. Finally, we distinguished thyroid dysfunction and thyroiditis to analyze the association of each disease with thyroid cancer, as these have not been differentiated in most other studies.

There are also several limitations. First, since thyroid ultrasonography for screening purposes is not covered by insurance and is not included in the NHIS dataset, we could not adjust for the frequency of thyroid ultrasonography tests. Instead, we used the number of thyroid function tests, which is the most representative workup in patients with thyroid dysfunction and thyroiditis. Second, there was no information on the treatment and severity of thyroid disease. In particular, patients with diagnoses of both hyperthyroidism and hypothyroidism may have been treated for hyperthyroidism in the past or may have developed hypothyroidism after the thyrotoxic phase of thyroiditis. There were patients with coexisting thyroid diseases such as thyroid dysfunction and thyroiditis. Therefore, we analyzed thyroid diseases by adjusting for other thyroid diseases in a logistic regression model and controlled for these effects in the subgroup analyses.

## 5. Conclusions

The screening effect likely significantly contributed to the positive associations of thyroid dysfunction and thyroiditis with thyroid cancer. After adjustment for this confounder, thyroid cancer risk was positively associated, but weaker, with hypothyroidism and thyroiditis, and negatively associated with hyperthyroidism and Graves’ disease. Our results suggest that over-screening for thyroid cancer may be reduced in patients with thyroid dysfunction and thyroiditis, particularly in hyperthyroidism and Graves’ disease.

## Figures and Tables

**Figure 1 cancers-13-05385-f001:**
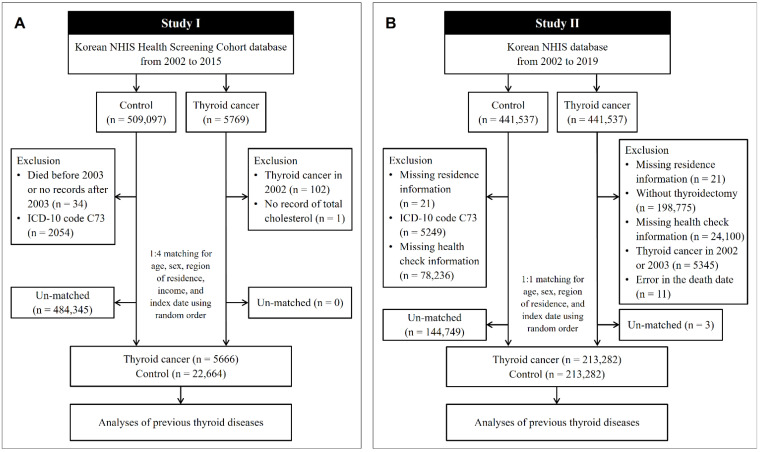
A schematic illustration of the participant selection process used in the present study. (**A**) In Study I, 5666 thyroid cancer patients were 1:4 matched with 22,664 control participants for age, sex, region of residence, and income. (**B**) In Study II, 213,282 thyroid cancer patients were 1:1 matched with 213,282 control participants for age, sex, and region of residence. NHIS, National Health Insurance Service; ICD, International Classification of Disease.

**Figure 2 cancers-13-05385-f002:**
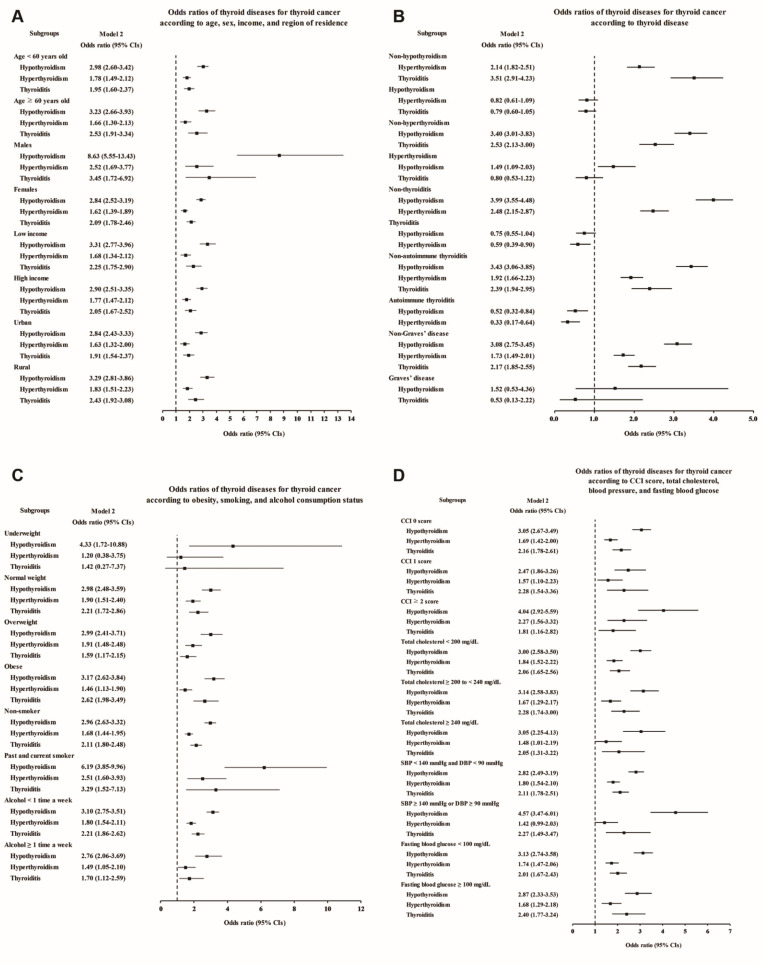
Adjusted odds ratios for thyroid cancer in the subgroup analyses of Study I. The subgroup analyses by (**A**) age, sex, income, and region of residence, (**B**) hypothyroidism, hyperthyroidism, thyroiditis, autoimmune thyroiditis, and Graves’ disease, (**C**) obesity by BMI, smoking status, and frequency of alcohol intake, and (**D**) CCI, total cholesterol, SBP, DBP, and fasting blood glucose. CCI, Charlson comorbidity index; SBP, systolic blood pressure; DBP, diastolic blood pressure.

**Figure 3 cancers-13-05385-f003:**
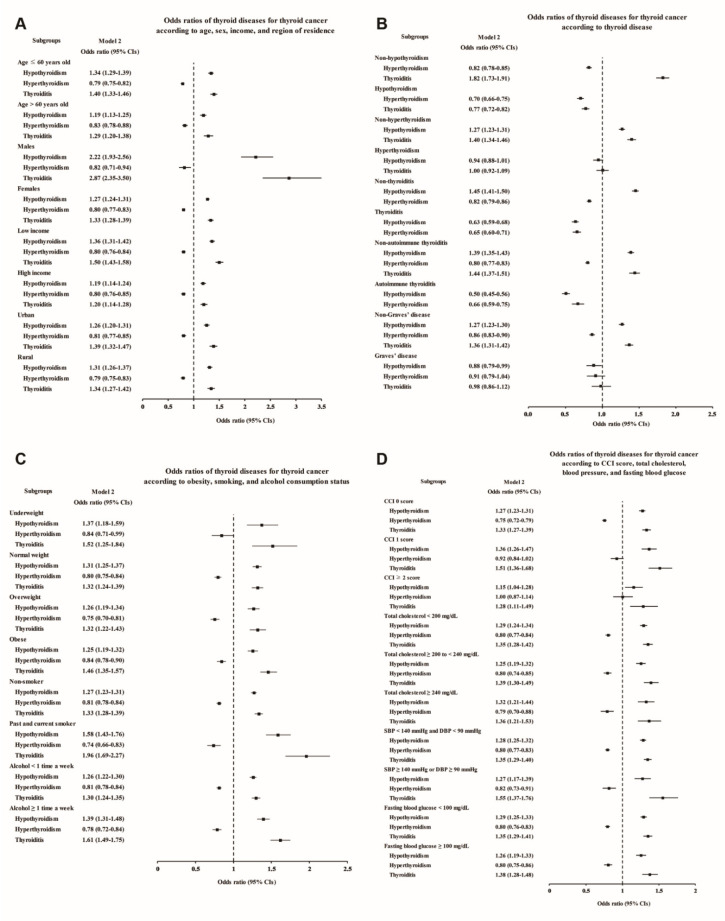
Adjusted odds ratios for thyroid cancer in the subgroup analyses of Study II. The subgroup analyses by (**A**) age, sex, income, and region of residence, (**B**) hypothyroidism, hyperthyroidism, thyroiditis, autoimmune thyroiditis, and Graves’ disease, (**C**) obesity by BMI, smoking status, and frequency of alcohol intake, and (**D**) CCI, total cholesterol, SBP, DBP, and fasting blood glucose. CCI, Charlson comorbidity index; SBP, systolic blood pressure; DBP, diastolic blood pressure.

**Table 1 cancers-13-05385-t001:** General characteristics of participants of Study I.

		Study I		
Characteristics		Thyroid Cancer	Control	*p*-Value
Age (years old, *n*, %)				1.000
	40–44	122 (2.2)	488 (2.2)	
	45–49	816 (14.4)	3264 (14.4)	
	50–54	1551 (27.4)	6204 (27.4)	
	55–59	1315 (23.2)	5260 (23.2)	
	60–64	876 (15.5)	3504 (15.5)	
	65–69	564 (10.0)	2256 (10.0)	
	70–74	308 (5.4)	1232 (5.4)	
	75–79	89 (1.6)	356 (1.6)	
	80–84	24 (0.4)	96 (0.4)	
	85+	1 (0.0)	4 (0.0)	
Sex (*n*, %)				1.000
	Males	1196 (21.1)	4784 (21.1)	
	Females	4470 (78.9)	17,880 (78.9)	
Region of residence (*n*, %)				1.000
	Urban	2719 (48.0)	10,876 (48.0)	
	Rural	2947 (52.0)	11,788 (52.0)	
Income (*n*, %)				1.000
	1 (lowest)	708 (12.5)	2832 (12.5)	
	2	664 (11.7)	2656 (11.7)	
	3	871 (15.4)	3484 (15.4)	
	4	1151 (20.3)	4604 (20.3)	
	5 (highest)	2272 (40.1)	9088 (40.1)	
Total cholesterol (mg/dL, mean, SD)		199.1 (37.8)	201.6 (37.2)	<0.001 †
SBP (mmHg, mean, SD)		124.3 (15.9)	123.4 (16.2)	<0.001 †
DBP (mmHg, mean, SD)		77.5 (10.5)	76.7 (10.6)	<0.001 †
Fasting blood glucose (mg/dL, mean, SD)		97.2 (23.0)	97.4 (24.7)	0.206
Obesity (*n*, %) ‡				<0.001 *
	Underweight	76 (1.3)	517 (2.3)	
	Normal weight	1872 (33.0)	8388 (37.0)	
	Overweight	1611 (28.4)	6207 (27.4)	
	Obese I	1870 (33.0)	6804 (30.0)	
	Obese II	237 (4.2)	748 (3.3)	
Smoking status (*n*, %)				<0.001 *
	Nonsmoker	4959 (87.5)	19,417 (85.7)	
	Past smoker	386 (6.8)	1413 (6.2)	
	Current smoker	321 (5.7)	1834 (8.1)	
Alcohol consumption (*n*, %)				0.520
	<1 time a week	4430 (78.2)	17,630 (77.8)	
	≥1 time a week	1236 (21.8)	5034 (22.2)	
CCI score (score, *n*, %) §				
	0	3441 (60.7)	17,657 (77.9)	<0.001 *
	1	918 (16.2)	2750 (12.1)	
	≥2	1307 (23.1)	2257 (10.0)	
Thyroid diseases (*n*, %)				
	Hypothyroidism	710 (12.5)	883 (3.9)	<0.001 *
	Hyperthyroidism	369 (6.5)	662 (2.9)	<0.001 *
	Thyroiditis	339 (6.0)	445 (2.0)	<0.001 *
	Autoimmune thyroiditis	150 (2.7)	205 (0.9)	<0.001 *
	Graves’ disease	41 (0.7)	77 (0.3)	<0.001 *

* Chi-square test. Significance at *p* < 0.05; † Wilcoxon rank-sum test. Significance at *p* < 0.05; ‡ Obesity (body mass index, kg/m^2^) was categorized as <18.5 (underweight), ≥18.5 to <23 (normal weight), ≥23 to <25 (overweight), ≥25 to <30 (obese I), and ≥30 (obese II); § CCI score was calculated without thyroid cancer; SD, standard deviation; SBP, systolic blood pressure; DBP, diastolic blood pressure; CCI, Charlson comorbidity index.

**Table 2 cancers-13-05385-t002:** Odds ratios (95% confidence interval) for thyroid cancer in hypothyroidism, hyperthyroidism, thyroiditis, autoimmune thyroiditis, and Graves’ disease.

	Thyroid Cancer	Control								
	(Exposure/Total, %)	(Exposure/Total, %)	Crude	*p*-Value	Model 1 ‡	*p*-Value	Model 2 §	*p*-Value	Model 3 ‖	*p*-Value
Study I †										
Hypothyroidism	710/5666 (12.5)	883/22,664 (3.9)	3.59 (3.23–3.98)	<0.001 *	3.65 (3.28–4.07)	<0.001 *	3.05 (2.73–3.41)	<0.001 *	3.44 (3.08–3.84)	<0.001 *
Hyperthyroidism	369/5666 (6.5)	662/22,664 (2.9)	2.32 (2.03–2.64)	<0.001 *	2.32 (2.03–2.66)	<0.001 *	1.73 (1.50–2.00)	<0.001 *	N/A	
Thyroiditis	339/5666 (6.0)	445/22,664 (2.0)	3.21 (2.77–3.71)	<0.001 *	3.17 (2.73–3.69)	<0.001 *	2.12 (1.81–2.49)	<0.001 *	N/A	
Autoimmune thyroiditis	150/5666 (2.7)	205/22,664 (0.9)	2.99 (2.42–3.70)	<0.001 *	2.96 (2.38–3.70)	<0.001 *	N/A		1.79 (1.42–2.26)	<0.001 *
Graves’ disease	41/5666 (0.7)	77/22,664 (0.3)	2.14 (1.46–3.13)	<0.001 *	2.15 (1.45–3.19)	<0.001 *	N/A		1.76 (1.17–2.66)	0.007 *
Study II										
Hypothyroidism	25,466/213,282 (11.9)	8994/213,282 (4.2)	3.08 (3.00–3.16)	<0.001 *	1.30 (1.26–1.33)	<0.001 *	1.28 (1.25–1.32)	<0.001 *	1.30 (1.26–1.34)	<0.001 *
Hyperthyroidism	14,707/213,282 (6.9)	6482/213,282 (3.0)	2.36 (2.29–2.43)	<0.001 *	0.84 (0.81–0.87)	<0.001 *	0.80 (0.77–0.83)	<0.001 *	N/A	
Thyroiditis	14,141/213,282 (6.6)	4647/213,282 (2.2)	3.19 (3.08–3.30)	<0.001 *	1.39 (1.33–1.44)	<0.001 *	1.36 (1.31–1.42)	<0.001 *	N/A	
Autoimmune thyroiditis	6562/213,282 (3.1)	2022/213,282 (1.0)	3.32 (3.15–3.49)	<0.001 *	1.22 (1.15–1.29)	<0.001 *	N/A		1.17 (1.11–1.24)	<0.001 *
Graves’ disease	5304/213,282 (2.5)	2207/213,282 (1.0)	2.44 (2.32–2.56)	<0.001 *	0.72 (0.68–0.76)	<0.001 *	N/A		0.69 (0.65–0.74)	<0.001 *

* Conditional logistic regression for Study I; Unconditional logistic regression for Study II; Significance at *p* < 0.05; † Models of Study I were stratified by age, sex, region of residence, and income; ‡ Model 1 of Study I was adjusted for total cholesterol, systolic blood pressure, diastolic blood pressure, fasting blood glucose, obesity, smoking, alcohol consumption, and Charlson comorbidity index scores; Model 1 of Study II was adjusted for the covariates in that of Study I plus age, sex, region of residence, income, and the number of thyroid function tests; § Model 2 was adjusted for model 1 plus hypothyroidism, hyperthyroidism, and thyroiditis; ‖ Model 3 was adjusted for model 1 plus hypothyroidism, autoimmune thyroiditis, and Graves’ disease.

**Table 3 cancers-13-05385-t003:** General characteristics of participants of Study II.

Characteristics		Study II		
		Thyroid Cancer	Control	*p*-Value
Age (years old, *n*, %)				1.000
	21–25	187 (0.1)	187 (0.1)	
	26–30	2186 (1.0)	2186 (1.0)	
	31–35	6600 (3.1)	6600 (3.1)	
	36–40	14,815 (7.0)	14,815 (7.0)	
	41–45	22,105 (10.4)	22,105 (10.4)	
	46–50	31,407 (14.7)	31,407 (14.7)	
	51–55	32,405 (15.2)	32,405 (15.2)	
	56–60	37,411 (17.5)	37,411 (17.5)	
	61–65	29,794 (14.0)	29,794 (14.0)	
	66–70	17,291 (8.1)	17,291 (8.1)	
	71–75	10,631 (5.0)	10,631 (5.0)	
	76+	8450 (4.0)	8450 (4.0)	
Sex (*n*, %)				1.000
	Males	37,527 (17.6)	37,527 (17.6)	
	Females	175,755 (82.4)	175,755 (82.4)	
Region of residence (*n*, %)				1.000
	Urban	98,967 (46.4)	98,967 (46.4)	
	Rural	114,315 (53.6)	114,315 (53.6)	
Income (*n*, %)				<0.001 *
	1 (lowest)	38,119 (17.9)	47,296 (22.2)	
	2	37,594 (17.6)	44,714 (21.0)	
	3	40,726 (19.1)	42,272 (19.8)	
	4	42,557 (20.0)	38,068 (17.9)	
	5 (highest)	54,286 (25.5)	40,932 (19.2)	
Total cholesterol (mg/dL, mean, SD)		194.5 (38.3)	196.5 (39.7)	<0.001 †
SBP (mmHg, mean, SD)		120.7 (15.0)	120.1 (15.5)	<0.001 †
DBP (mmHg, mean, SD)		75.4 (10.2)	74.9 (10.3)	<0.001 †
Fasting blood glucose (mg/dL, mean, SD)		96.1 (20.7)	96.4 (23.3)	<0.001 †
Obesity (*n*, %) ‡				<0.001 *
	Underweight	6637 (3.1)	9364 (4.4)	
	Normal weight	84,051 (39.4)	92,615 (43.4)	
	Overweight	49,788 (23.3)	47,523 (22.3)	
	Obese I	61,825 (29.0)	54,812 (25.7)	
	Obese II	10,981 (5.2)	8968 (4.2)	
Smoking status (*n*, %)				<0.001 *
	Nonsmoker	180,976 (84.9)	175,046 (82.1)	
	Past smoker	14,407 (6.8)	12,858 (6.0)	
	Current smoker	17,899 (8.4)	25,378 (11.9)	
Alcohol consumption (*n*, %)				<0.001 *
	<1 time a week	150,844 (70.7)	145,881 (68.4)	
	≥1 time a week	62,438 (29.3)	67,401 (31.6)	
CCI score (score, *n*, %) §				<0.001 *
	0	168,375 (78.9)	171,442 (80.4)	
	1	34,820 (16.3)	23,235 (10.9)	
	≥2	10,087 (4.7)	18,605 (8.7)	
No. of thyroid function test		1.8 (2.6)	0.5 (1.5)	<0.001 †
Thyroid diseases (*n*, %)				
	Hypothyroidism	25,466 (11.9)	8994 (4.2)	<0.001 *
	Hyperthyroidism	14,707 (6.9)	6482 (3.0)	<0.001 *
	Thyroiditis	14,141 (6.6)	4647 (2.2)	<0.001 *
	Autoimmune Thyroiditis	6562 (3.1)	2022 (1.0)	<0.001 *
	Graves’ disease	5304 (2.5)	2207 (1.0)	<0.001 *

* Chi-square test. Significance at *p* < 0.05; † Wilcoxon rank-sum test. Significance at *p* < 0.05; ‡ Obesity (body mass index, kg/m^2^) was categorized as <18.5 (underweight), ≥18.5 to <23 (normal weight), ≥23 to <25 (overweight), ≥25 to <30 (obese I), and ≥30 (obese II); § CCI score was calculated without thyroid cancer; SD, standard deviation; SBP, systolic blood pressure; DBP, diastolic blood pressure; CCI, Charlson comorbidity index.

## Data Availability

The data presented in this study are available upon request from the corresponding author.

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
