# Peer review of "Screening Leads to Overestimated Associations of Thyroid Dysfunction and Thyroiditis with Thyroid Cancer Risk"

_cancers, 2021, doi:10.3390/cancers13215385_

Round 1

Reviewer 1 Report

Thank you for the opportunity to review your paper.

This is a very large epidemiology study about correlation between thyroid cancer and thyroid dysfunction. You have compared 2 groups not very homogeneus . In the group 2 the incidence of thyroid cancer is 11 times more than the  controll group. How to explain this diffence? Can effect the results ?

I think that is necessary clarify better what  do you mean for hypothyroidism,

 hyperthyroidism and thyroiditis. These conditions can coexit in the  same patients .  To identify this patients  with just  a codes is a bias for the results.It Is very strange a such large number of hypotyroidism in the Korean population. What are the reason? probably these patients have had a thyroiditis before.

Author Response

Point 1: This is a very large epidemiology study about correlation between thyroid cancer and thyroid dysfunction. You have compared 2 groups not very homogeneous. In the group 2 the incidence of thyroid cancer is 11 times more than the control group. How to explain this difference? Can effect the results?

Response 1: In Study I, thyroid cancer patients and control participants were 1:4 matched using the NHIS cohort data (n = 5,666 and 22,664, respectively), and in Study II, they were 1:1 matched in Study II using the entire NHIS data (n = 213,282, both), and their past history of thyroid diseases was analyzed. Table 2 shows the proportion of participants with previous thyroid diseases, such as hypothyroidism, hyperthyroidism, thyroiditis, autoimmune thyroiditis, and Graves’ disease, which we defined as “exposure”, in thyroid cancer and control groups. Thus, although the number of previous thyroid disease (exposure) itself was higher in Study II with a much larger data size, there were no significant differences in the ratios of exposure and the crude odd ratios between Study I and Study II. 

Point 2: I think that is necessary clarify better what do you mean for hypothyroidism, hyperthyroidism and thyroiditis. These conditions can coexit in the same patients. To identify this patients with just a codes is a bias for the results. It is very strange a such large number of hypotyroidism in the Korean population. What are the reason? probably these patients have had a thyroiditis before.

Response 2: We appreciate the precious comment and agreed with it. As we mentioned in Introduction section, previous studies often did not distinguish between thyroid dysfunction and thyroiditis. However, to exclude the effect of each disease in subjects with coexisting thyroid diseases, we analyzed thyroid diseases by adjusting for other thyroid disease in the logistic regression models and controlled for these effects in the subgroup analyses. In response to this comment, we have clearly mentioned this point in the Discussion section, as below.

[Discussion section, page 14]
There were patients with coexisting thyroid diseases such as thyroid dysfunction and thyroiditis. Therefore, we analyzed thyroid diseases by adjusting for other thyroid diseases in a logistic regression model and controlled for these effects in the subgroup analyses.

Reviewer 2 Report

The study by Song et al. determione the connection between functional thyroid disease ad thyroiditis and thyroid cancer. The interest in this topic has been going on for many years. The Authors showed that thyroid cancer risk is positively associated with hypothyroidism and thyroiditis but negatively associated with hyperthyroidism and Graves’ disease. Importantly, these studies were performed on on a large cohort of Korean National Health Insurance Service-Health Screening Cohort data. The manuscript is well-written and can be accepted in present form. I would only suggest correction of some typos that are listed below.

  1. Page 6, Footer in Table 1: I am not sure if it is correct to put the period right before the comma, i.e., ".;".
  2. Page 14: Please delete one extra period that is at the end of the Supplementary Materials section.
  3. Page 14, Funding section: “Please add:” should be removed.

Author Response

Point:
Page 6, Footer in Table 1: I am not sure if it is correct to put the period right before the comma, i.e., ".;".
Page 14: Please delete one extra period that is at the end of the Supplementary Materials section.
Page 14, Funding section: “Please add:” should be removed

Response: Thank you for the careful review. We corrected all typos pointed out by the reviewer.

Reviewer 3 Report

Comments to Authors.

Song et al. present an original article in which they have analyzed a relationship between thyroid dysfunction and a thyroid cancer (TC)  risk. I found that the study is well-designed as a nationwide, large population-based project.

A few doubts about this study have concerned me.

  1. Given the current study results Authors concluded that screening test for TC should be reduced in patients with thyroid dysfunction or thyroiditis. But, they should unify their conclusions. The section 5 [Conclusions] of the main text sounds differently than conclusions in the simple summary or in an abstract of the study, where conclusions look like more wide than the study results indicate.
  2. Taking into account that the present study is conducted as an element ‘from basic to clinical studies’ Authors could more clearly highlight the usefulness of their study in a routine clinical practice. My the most crucial clinical inquiry is: how to implement these results in the patients with thyroid disease? In which patients should we limit an assessment a risk of TC? In all the patients as the majority of the patients in endocrinological out-patient clinic have thyroid dysfunction or thyroiditis?

Author Response

Point 1: Given the current study results Authors concluded that screening test for TC should be reduced in patients with thyroid dysfunction or thyroiditis. But, they should unify their conclusions. The section 5 [Conclusions] of the main text sounds differently than conclusions in the simple summary or in an abstract of the study, where conclusions look like more wide than the study results indicate.

Response 1: We appreciate the reviewer’s thoughtful comment. As suggested by the reviewer, we now revised the section 5. Conclusions of the main text, and more clearly highlighted the usefulness of this study in the clinical practice. 

Point 2: Taking into account that the present study is conducted as an element ‘from basic to clinical studies’ Authors could more clearly highlight the usefulness of their study in a routine clinical practice. My the most crucial clinical inquiry is: how to implement these results in the patients with thyroid disease? In which patients should we limit an assessment a risk of TC? In all the patients as the majority of the patients in endocrinological out-patient clinic have thyroid dysfunction or thyroiditis?

Response 2: In our study, after adjustment for the screening effect, thyroid cancer risk was positively associated but weaker with hypothyroidism and thyroiditis, and negatively associated with hyperthyroidism and Graves’ disease. Because patients with thyroid disease had the modest increase or decrease in risk for thyroid cancer compared to controls, they would not need over-screening for thyroid cancer beyond the general population. However, especially in patients with hyperthyroidism and Graves’ disease, who have a reduced risk of thyroid cancer, excessive concern or routine assessment of thyroid cancer may be unnecessary. This is now described in the Conclusions section. 

[Conclusion section, page 14]
5. Conclusions
   The screening effect likely significantly contributed to the positive associations of thyroid dysfunction and thyroiditis with thyroid cancer. After adjustment for this confounder, thyroid cancer risk was positively associated but weaker with hypothyroidism and thyroiditis, and negatively associated with hyperthyroidism and Graves’ disease. Our results suggest that over-screening for thyroid cancer may be reduced in patients with thyroid dysfunction and thyroiditis, particularly in hyperthyroidism and Graves’ disease.